# Analyzing the Tourism–Energy–Growth Nexus for the Top 10 Most-Visited Countries

**Cem Işik [1],\*, Eyüp Doğan [2] and Serdar Ongan [3]**

[1]   Department of Tourism Management, Atatürk University, 25240 Erzurum, Turkey; isikc@atauni.edu.tr
[2]   Department of Economics, Abdullah Gül University, 38080 Kayseri, Turkey; eyup.dogan@agu.edu.tr
[3]   Department of Economics, St. Mary's College of Maryland, St. Mary's City, MD 20686, USA; songan@smscm.edu
\*   Correspondence: isikc@atauni.edu.tr; Tel.: +90 442 231 5043

**Abstract:** By using the Emirmahmutoglu–Kose bootstrap Granger non-causality method, this study explores the directions of causality among tourist arrivals, tourism receipts, energy consumption and economic growth for the top 10 most-visited countries (France, the USA, Spain, China, Italy, Turkey, Germany, the United Kingdom, Russia, and Mexico) in the world. This study finds a variety of causal directions between the pair of analyzed variables for each country and the panel. Since cross-sectional dependence exists across the top countries for the analyzed variables, the bootstrap Granger causality test that accounts for the mentioned issue in the estimation process presumably produces reliable and accurate outputs. Further results and policy implications are discussed in this empirical study.

**Keywords:** Granger causality; tourist arrivals; tourism receipts; energy consumption; economic growth; the top 10 most-visited countries

## 1. Introduction

A general idea has emerged in the last decade: that tourism flows and energy consumption increase revenue, produce growth, create employment in tourism and energy sectors and cause a general improvement in economic development (Crouch and Ritchie 1999). Currently, both energy and tourism are related to sustainability of the economy in the world. (Daly 1991; Hall and Page 2014). According to the U.S. Energy Information Administration (EIA), world consumption of marketed energy will increase from 549 quadrillion British thermal units (Btu) in 2012 to 815 quadrillion Btu in 2030 (EIA 2016). Similarly, the United Nations World Tourism Organization (UNWTO) projects that world international tourist arrivals will increase from 1.0 billion in 2012 to 1.8 billion in 2030 (UNWTO 2013). Especially in developing countries, tourism flows and energy consumption can impact positively on the trade balance, employment and limited resources of host countries. (Daly 1991; Asif and Muneer 2007; Isik 2012, 2013; Hall and Page 2014; Dogan et al. 2015; Isik, 2015; Isik and Shahbaz 2015; Dogan and Seker 2016; Ertugrul et al. 2016; Isik et al., 2017b, 2017a).

With the continual increase in tourism and travel activities globally, there are serious allegations that the industry is significantly contributing towards climate change through its impact on CO2 emissions (Sharpley and Telfer 2015). According to Karabuğa et al. (2015), 90% of energy consumption occurs during outgoing and incoming to destinations (43% airway, 42% land transport, 15% sea and railways). Tourism sector has a 5% of worldwide carbon dioxide (CO2) emission. These impacts might be reduced if proper courses of actions are to be taken (Dogru et al. 2016) The current literature recommends

reducing the consumption of traditional energy resources while increasing the use of renewable or alternative energy for sustainable economic growth and tourism development (Scott and Becken 2010; Jenkins and Nicholls 2010). Renewable energy resources (Solar Energy, Biomass Energy, Heat Pump, Wind Power and Geothermal Energy etc.) are the most suitable energy forms for clean environment concept that do not pollute during the production and renew it (Karabuğa et al. 2015).

The developed countries that are the main generators of tourists, much more attention has been given to tourism development theory in the context of the less developed world. Nevertheless, the developed countries relatively reach major benefits from the tourism. While less developed countries' contribution to energy due to tourism activities is smaller compared to developed countries (Sharpley and Telfer 2015). In especial, the tourism sector has shown an exceptional improvement in France, the USA, Spain, China, Italy, Turkey, Germany, the United Kingdom, Russia, and Mexico, which are the top ten most-visited countries in terms of tourist arrivals in 2014. Table 1 shows tourist arrivals, tourism receipt, energy consumption and GDP statistics for the top 10 most-visited countries in the world for the latest available year.

**Table 1.** Tourist Arrivals, Tourism Receipt, Energy Consumption and GDP of the Top 10 most-Visited Countries.

| Top 10 Most-Visited Countries (2014) | Int. Tourist Arrivals (Million People Ranking, 2014) | Int. Tourism Receipt (Billion USD, 2014) | Energy Consumption (Quadrillion Btu, 2013) | GDP (2014) (The Real Gross Domestic Product Constant in 2005 USD) |
|---|---|---|---|---|
| France | 83.7 (1) | 55.4 | 10.7 | 2,829,192 |
| The USA | 74.8 (2) | 177.2 | 95.05 | 17,419,000 |
| Spain | 66.0 (3) | 65.2 | 6.02 | 1,381,342 |
| China | 55.6 (4) | 56.9 | 105.88 | 10,354,832 |
| Italy | 48.6 (5) | 45.5 | 7.17 | 2,141,161 |
| Turkey | 39.8 (6) | 43.3 | 5.05 | 798,429 |
| Germany | 33.0 (7) | 43.3 | 13,.47 | 3,868,291 |
| The UK | 32.6 (8) | 45.3 | 8.63 | 2,988,893 |
| Russia | 29.8 (9) | 11.7 | 31.52 | 1,860,598 |
| Mexico | 29.1 (10) | 17,4 | 7.75 | 1,294,690 |
| Total | 493 | 561.2 | 291.24 | 44,936,428 |
| World | 1113 | 1245 | 524.08 | 77,609 Trillion USD |
| Top 10's share | 44% | 45% | 56% | 58% |

( ) denotes the ranks of the countries. Source: World Bank, World Development Indicators (WDI 2016). UNWTO (2013), (United Nations World Tourism Organization), World Tourism Trends. British Petroleum Energy Outlook (2015). EIA (2016), International Energy Agency.

According to the World Development Indicators (WDI) (2016), the top 10 most-visited countries (France, the USA, Spain, China, Italy, Turkey, Germany, the United Kingdom, Russia, and Mexico) reached 44.936 billion USD GDP in 2014 (with a share of 58% of total world GDP). These countries total trade was close to 1.657 trillion USD, exports were 7.828 billion USD and imports were 8.739 billion USD. The top 10 most-visited countries' economies reached an annual average growth rate of 2.00% as of 2014. The tourism receipt for the top 10 most-visited countries was 561.2 billion USD in 2014 and 660.8 billion USD in 2013. The international tourism receipt for the analyzed countries reached an annual average growth rate of 18% as of 2014. Energy consumption in these countries is increasingly growing in the last decade and reached 524.08 quadrillion Btu (British thermal unit) in 2013. It was more than half of the world's energy consumption (56% in 2013). The top 10 countries' energy consumption was 291.24 quadrillion Btu of the world total 524.08 quadrillion Btu in 2013 (British Petroleum Energy Outlook 2015).

The purpose of this study is to investigate the causal relationships between tourism development, energy consumption and real GDP (economic growth). For this purpose, the Bootstrap Panel Granger causality test is used to examine the causal link between tourism, energy consumption and real GDP, in France, the USA, Spain, China, Italy, Turkey, Germany, the United Kingdom, Russia, and Mexico. The top 10 most-visited countries offer a unique setting to investigate the causal relationship between tourism development, economic growth, and energy consumption because of their respective shares of world's tourism development, GDP and energy consumption.

This research contributes to the existing literature in several aspects. The investigation of the top countries is of interest to policy makers and governments as they play important roles in energy and tourism sectors and in the overall world economy upon aforementioned discussions. This study uses the recently developed Emirmahmutoglu–Kose bootstrap panel Granger causality test which accounts for the issue of cross-sectional dependence, since we find that the issue appears in the analyzed data. The reported results are thus reliable and robust, and strong for policy implications.

The rest of this paper is systematized as follows. Section 2 presents the main findings of the previous studies on this nexus, Section 3 defines the methodology, Section 4 discusses the empirical results and finally, Section 5 elaborates upon the conclusions and policy recommendations.

## 2. Literature Review

The tourism and energy consumption and their relation with economic growth is not involved in many studies until recent years. Some early works have focused on tourism economics or energy economics itself. Few studies use these two variables which affect economic growth in one equation. The impact of the tourism and energy consumption on economic growth has not always been well argued in the economic literature.

It is appropriate for our study to begin investigating the literature that has stressed the link between the tourism-energy-growth. There are some earlier studies on energy consumption, the tourism sector and economic growth (Oh et al. 2010; Akbostancı et al. 2011; Liu et al. 2011; O'Mahony et al. 2012; Pardo et al. 2012; Pace 2015; Moutinho et al. 2015; Isik et al., 2017b, 2017a). We also discovered different works in the tourism literature that have examined the energy and $CO_2$ emissions (Liu et al. 2011; Scott 2011; Wu and Shi 2011; Lee and Brahmasrene 2013; Lee and Kwag 2013; Katircioglu et al. 2014).

The actual literature focused on the energy-growth relation displays a wide variety of result. Today, the energy—growth connection has widely been empirically examined since the study of Kraft and Kraft. The general results of the current works on energy consumption are not uniform. Some different econometric technics were used (unit root tests, cointegration tests, etc.) to identify the causality direction between these variables (Shahbaz and Lean 2012).

We have also studied the main findings of the economic literature for the impact of energy consumption and tourism flows on economic growth as shown in Table 2 (Energy, Tourism Economic Growth Causality). Based on the results, we can classify the research into four acceptable theories; growth, conservation, feedback and neutrality theory.

**Table 2.** Energy, Tourism and Economic Growth Causality.

| Author | Time | Destination | Methodology | Variables | Causality |
|---|---|---|---|---|---|
| | | | **Energy/Tourism—Led Growth** | | |
| Isik et al. (2017a) | 1970–2014 | Greece | Zivot Andrews unit rooti ARDL, Granger Causality | ARRV/RCPT&Y&CO2&FD&I | ARRV/RCPT $\rightarrow$ CO2; FD $\rightarrow$ CO2, I $\rightarrow$ CO2; Y $\rightarrow$ CO2 |
| Ozturk (2016) | | The panel of selected 34 countries | FMOLS | ARRV/RCPT&Y&EGY | EGY exerts a negative association with the ARRV/RCPT |
| Antonakakis et al. (2015) | 1995–2012 | Greece, Italy, Portugal, Spain, Australia, Germany. | Spillover index approach | ARRV/RCPT&Y | $\backsimeq$ for Australia and Greece ARRV/RCPT $\rightarrow$ Y for Italy, Portugal, Spain and Germany |
| Dogan (2015) | 1990–2012 | Turkey | ARDL | EGY & Y | EGY $\backsimeq$ Y in the short run EGY $\rightarrow$ Y in the long run |
| Rezitis and Ahammad (2015) | 1990–2012 | South and Southeast Asian Countries | Panel Unit Root LLC and Breitung Tests, Panel Unit Root HT and IPS, Panel Co-integration: Panel FMOLS and DOLS | EGY & Y | EGY $\rightarrow$ Y |
| Dogan (2014) | 1971–2011 | Sub-Saharan Africa | ADF, Johansen ML Co-integration, Granger Causality | EGY & Y | EGY $\rightarrow$ Y in Kenya $\backsimeq$ in Benin, Congo and Zimbabwe |

**Table 2.** *Cont.*

| Author | Time | Destination | Methodology | Variables | Causality |
|---|---|---|---|---|---|
| **Energy/Tourism—Led Growth** | | | | | |
| Leon et al. (2014) | 1998–2006 | 14 Developed, 31 less developed | The Generalized Method of Moments, GLS | ARRV/RCPT &Y&$CO_2$&Population&E | Tourism has pozitive effect on 14 developed and 31 less developed countries |
| Adhikari and Chen (2013) | 1990–2009 | 80 Developing countries | Panel Unit Root, Panel Cointegration and Panel Dynamic Ordinary Least Squares (DOLS) | EGY & Y | EGY →Y |
| Kareem (2013) | 1990–2011 | Africa | Levin, Lin and Chu, Im, Pesaran and Shin Hadri Z, Pedroni | ARRV/RCPT &Y&others | ARRV/RCPT → Y |
| Isik (2010) | 1977–2008 | Turkey | ARDL | EGY & Y | EGY → Y in the short run |
| Cortes-Jimenez and Pulina (2010) | 1954–2000 | Italy & Spain | VECM, Johansen Cointegration | ARRV/RCPT &Y&C | ARRV/RCPT → Y |
| **Growth—Led Energy/Tourism** | | | | | |
| Isik et al. (2017a) | | Top 7 | ADF, PP, Nonlinear Unit Root, Panel Co-integration, Bootstrap | ARRV/RCPT&Y&EGY | EGY → Y (Spain); ARRV/RCPT → Y (China, Turkey); ARRV/RCPT → EGY (Italy, Spain, Turkey and United States) |
| Azam et al. (2015b) | 1980–2012 | Indonesia, Malaysia and Thailand | CSUSM and SUCUSM Park, ADF for Stationary: Least square | EGY & Y | Y → EGY (Indonesia, Malaysia and Thailand) |
| Azam et al. (2015a) | 1980–2012 | Indonesia, Malaysia, Philippines, Singapore and Thailand | ADF Unit Root, Cointegration, Pearson Correlation Analysis, Granger Causality | EGY & Y | Y → EGY (Malaysia) ≏ (Indonesia, Philippines, Singapore and Thailand) |
| He and Zheng (2011) | 1990–2009 | China (Sichuan) | VECM | ARRV/RCPT&Y | Y → ARRV/RCPT |
| Ozturk et al. (2010) | 1971–2005 | 51 Low and middle income countries | IPS Panel Unit Root, Panel Co-integration, Panel Granger Causality, Panel FMOLS and DOLS | EGY & Y | Y → EGY (low income countries) EGY ↔ Y (middle income countries) |
| **No Causality** | | | | | |
| Jin (2011) | 1974–2004 | Hong Kong | VECM, Variance Decompositions (VDCs) | ARRV/RCPT&Y | ≏ |
| **Bidirectional Causality** | | | | | |
| Dogru and Bulut (2017) | 1996–2014 | 7 EU Countries | CADF panel unit root tes | ARRV/RCPT&Y | Y ↔ ARRV/RCPT |
| Al-mulali et al. (2014) | 1985–2012 | Middle East | Pedroni Cointegration/VECM | ARRV/RCPT &Y&Real Exchange Rate&Total Trade | Y ↔ ARRV/RCPT |
| Tang and Abosedra (2014) | 2001–2009 | MENA | Static Panel Estimation, Arellano-Bond Dynamic GMM Estimation | EGY, Y, ARRV/RCPT, Poliitical Stability, C | EGY ↔ Y |
| Lee and Brahmasrene (2013) | 1988–2005 | EU countries | Pearson Correlation Analysis, Unit Root, Johansen Panel Cointegration | ARRV/RCPT&Y E&$CO_2$&FDI | ARRV/RCPT → Y Y → $CO_2$, EGY Tourism does not cause an increase of $CO_2$ emmissions FDI → Y FDI cause a decrease of $CO_2$ emissions |
| Tiwari et al. (2013) | 1995–2005 | OECD | Panel VAR | ARRV/RCPT&EGY&$CO_2$ | EGY ↔ ARRV/RCPT |
| Kadir and Karim (2012) | 1998–2005 | ASEAN | Pedroni Cointegration/VECM | ARRV/RCPT&Y | Y ↔ ARRV/RCPT |

GDP = Y, EGY = Energy Consumption, ARRV/RCPT = Tourism, C = capital, $CO_2$ = Carbon Dioxide, I = International Trade, FD = Financial Development, Emission, ECM = Error Correction Model, JJ = Johansen-Juselius, VEC = Vector Error Correction Model, DOLS = The Panel Dynamic Ordinary Least Squares FMOLS = the Fully-Modified Ordinary Least Squares, GMM = The Dynamic Generalised Method of Moments, ARDL = Autoregressive Distributed Lag, GLS = Generalized Least Squares, DF = Dickey Fullers, LLC = Levin-Lin-Chu Test, IPS = The Im-Pesaran-Shin Test, HT = Harris–Tsavalis Test, CUSUM = Cumulative Sum, CUSUSQ = Cumulative Sum of Squares, PP = Phillips–Perron, KPSS = Kwiatkowski–Phillips–Schmidt–Shin, HEGY = the Hylleberg-Engle-Granger-Yoo, VDC = Variance Decompositions and →, ←, ↔, ≏ shows unidirectional causality, bidirectional causality, and no causality, respectively.

It is generally agreed that tourism and energy plays a robust role for both the income and the expenditure and investment of goods and services within an economy. In light of these findings,

the mission of this section is to ensure a review of the previous studies on the causal link between tourism, energy and growth. In the related literature, there have been more studies that have applied Granger causality tests to investigate the causal relationships for tourism development *or* energy consumption with economic growth as pairs. For instance, while Aqeel and Butt (2008) apply the Hsiao's Granger causality test for Pakistan, Wolde-Rufael (2004) applies the Toda and Yamamoto (1995) causality test and they both find causal relationships from energy consumption to the economic growth. However, Ozturk et al. (2010) apply the Panel Granger causality test on 51 low and middle-income countries and find the same relationship from the economic growth to energy consumption. Furthermore, while Chen et al. (2007) apply the Panel causality for China and find no causal relationships between energy consumption and economic growth, Yuan et al. (2007) apply the Granger causality for the same country and find bidirectional causality between these two variables.

## 3. Model and Data

Following studies Lee and Brahmasrene (2013), Tiwari et al. (2013), Leon et al. (2014), and Tang et al. (2016) that focus on the link of tourism-energy-growth nexus, this study uses the following models where economic growth ($Y$) is the dependent variable, and energy consumption (EGY) and tourism are the independent variables. We used E-Views 8 econometric software for the estimations. To check the robustness of the direction of causality between tourism and economic growth, we employ both tourist arrivals (ARRV) and tourism receipt (RCPT). The models can be written as:

$$\text{Model 1: } Y_{it} = f(\text{EGY}_{it}, \text{ARRV}_{it}/\text{RCPT}_{it})$$

where $t$ and $i$ denote the time period and country. According to the World Development Indicators (WDI) (2016), the world's 10 most-visited countries are France, the US, Spain, China, Italy, Turkey, Germany, the UK, Russia and Mexico.[1] Regarding data description, economic growth ($Y$) is measured by the real gross domestic product constant in 2005 USD; energy consumption (EGY) is shown in kg of oil equivalent; tourist arrival (ARRV) equals the number of international inbound tourists; tourism receipts (RCPT) are measured by expenditures by international inbound tourists constant in 2005 USD. The annual data for the analyzed variables from 1995–2013 are sourced from the WDI (2016). It is worth that we use available data with the longest time period.

## 4. Methods and Empirical Results

As it is the main research proposal of this study to investigate the directions of causality among economic growth, energy consumption and tourism, we should find an appropriate and reliable estimation technique. One of the commonly observed but ignored issues in the literature is the presence of cross-sectional dependence across countries for panel data. Besides, traditional Granger causality approaches such as pair-wise Granger causality test and Granger causality based on vector error correction mechanism, this may produce inconsistent output because they do not take into account the issue of cross-sectional dependence. In the existence of cross-sectional dependence, we should employ a second generation Granger causality method robust enough to handle this issue. To this end, we analyze whether or not the analyzed variables include cross-sectional dependence by using the Pesaran's cross-sectional dependence (CD) test (Pesaran 2004).

Results from the CD-test are reported in Table 3. We have enough evidence to reject the null hypothesis of cross-sectional independence in favor of the alternative hypothesis of cross-sectional dependence across the top 10 most-visited countries for economic growth, energy consumption, tourist arrivals and tourism receipts at 1% level of significance. This study, according to the reported results, would rather use the bootstrap methodology to Granger causality test for cross-sectionally dependent

---

[1] The data are available at http://data.worldbank.org/.

panels developed by Emirmahmutoglu and Kose (2011) than the above mentioned traditional ones. The Emirmahmutoglu–Kose bootstrap causality test builds on the Meta analysis of Fisher (Fisher 1932) and the idea of vector autoregression (VAR) with lag order and the maximal order of integration due to Toda and Yamamoto (1995). To test the null hypothesis of non-Granger causality, the authors estimated a level of VAR with lag order ($k_i$) and the maximum order of integration of variables ($dmax_i$) in heterogonous mixed panels.[2] The only prior information needed is $dmax_i$ suspected to happen in the system for each country.

**Table 3.** Cross-sectional independence test.

|  | Y | EGY | ARRV | RCPT |
|---|---|---|---|---|
| CD-test | 27.00 * | 7.18 * | 23.77 * | 5.56 * |
| *p*-value | 0.00 | 0.00 | 0.00 | 0.00 |

* denotes the statistical significance at 1% level.

By following the original study, we applied the Augmented Dickey–Fuller (ADF) unit root test (Dickey and Fuller, to the analyzed time-series data so as to determine the maximum number of integration). Moreover, we also used the Phillips-Perron unit root test for the purpose of robustness. Results from the ADF unit root test and the Phillips-Perron unit root tests are given in Tables 4 and 5, respectively. Both tests virtually produce the same order of integration of analyzed variables. Overall, the maximum order of integration (dmax) is determined to be two for each variable for the panel.

**Table 4.** Augmented Dickey–Fuller (ADF) unit root test.

|  | Y | | | | EGY | | | |
|---|---|---|---|---|---|---|---|---|
|  | Level | 1st difference | 2nd difference | $dmax_i$ | Level | 1st difference | 2nd difference | $dmax_i$ |
| China | 0.92 | 0.45 | 0.01 | 2 | 0.97 | 0.28 | 0.01 | 2 |
| France | 0.13 | 0.09 | 0.00 | 2 | 0.12 | 0.79 | 0.00 | 2 |
| Germany | 0.59 | 0.01 | - | 1 | 0.95 | 0.00 | - | 1 |
| Italy | 0.20 | 0.08 | 0.00 | 2 | 0.79 | 0.07 | 0.00 | 2 |
| Mexico | 0.34 | 0.02 | - | 1 | 0.76 | 0.00 | - | 1 |
| Russia | 0.93 | 0.02 | - | 1 | 0.91 | 0.01 | - | 1 |
| Spain | 0.16 | 0.73 | 0.00 | 2 | 0.15 | 0.43 | 0.00 | 2 |
| Turkey | 0.90 | 0.01 | - | 1 | 0.87 | 0.00 | - | 1 |
| UK | 0.12 | 0.16 | 0.00 | 2 | 0.99 | 0.00 | - | 1 |
| USA | 0.04 | - | - | 0 | 0.18 | 0.01 | - | 1 |
|  | ARRV | | | | RCPT | | | |
|  | Level | 1st difference | 2nd difference | $dmax_i$ | Level | 1st difference | 2nd difference | $dmax_i$ |
| China | 0.33 | 0.00 | - | 1 | 0.02 | 0.00 | - | 1 |
| France | 0.10 | 0.07 | 0.00 | 2 | 0.12 | 0.04 | - | 1 |
| Germany | 0.97 | 0.00 | - | 1 | 0.20 | 0.01 | - | 1 |
| Italy | 0.51 | 0.01 | - | 1 | 0.56 | 0.01 | - | 1 |
| Mexico | 0.77 | 0.00 | - | 1 | 0.16 | 0.03 | - | 1 |
| Russia | 0.01 | - | - | 0 | 0.04 | - | - | 0 |
| Spain | 0.15 | 0.14 | 0.00 | 2 | 0.05 | - | - | 0 |
| Turkey | 0.87 | 0.00 | - | 1 | 0.91 | 0.00 | - | 1 |
| UK | 0.87 | 0.06 | 0.00 | 2 | 0.98 | 0.07 | 0.00 | 2 |
| USA | 0.95 | 0.03 | - | 1 | 0.93 | 0.01 | - | 1 |

---

[2]   Please refer to the original study by Emirmahmutoglu and Kose (2011) for detailed information.

**Table 5.** Phillips–Perron unit root test.

| | Y | | | | EGY | | | |
|---|---|---|---|---|---|---|---|---|
| | Level | 1st difference | 2nd difference | dmax$_i$ | Level | 1st difference | 2nd difference | dmax$_i$ |
| China | 0.97 | 0.39 | 0.01 | 2 | 0.99 | 0.32 | 0.00 | 2 |
| France | 0.01 | - | - | 0 | 0.11 | 0.00 | - | 1 |
| Germany | 0.29 | 0.00 | - | 1 | 0.68 | 0.00 | - | 1 |
| Italy | 0.21 | 0.09 | 0.00 | 2 | 0.69 | 0.08 | 0.00 | 2 |
| Mexico | 0.36 | 0.02 | - | 1 | 0.74 | 0.00 | - | 1 |
| Russia | 0.93 | 0.02 | - | 1 | 0.93 | 0.00 | - | 1 |
| Spain | 0.12 | 0.76 | 0.00 | 2 | 0.35 | 0.44 | 0.00 | 2 |
| Turkey | 0.90 | 0.01 | - | 1 | 0.88 | 0.00 | - | 1 |
| UK | 0.12 | 0.17 | 0.00 | 2 | 0.95 | 0.00 | - | 1 |
| USA | 0.04 | - | - | 0 | 0.18 | 0.01 | - | 1 |
| | ARRV | | | | RCPT | | | |
| | Level | 1st difference | 2nd difference | dmax$_i$ | Level | 1st difference | 2nd difference | dmax$_i$ |
| China | 0.13 | 0.00 | - | 1 | 0.03 | 0.00 | - | 1 |
| France | 0.10 | 0.07 | 0.00 | 2 | 0.28 | 0.19 | 0.01 | 2 |
| Germany | 0.97 | 0.00 | - | 1 | 0.15 | 0.01 | - | 1 |
| Italy | 0.51 | 0.00 | - | 1 | 0.56 | 0.01 | - | 1 |
| Mexico | 0.83 | 0.00 | - | 1 | 0.30 | 0.03 | - | 1 |
| Russia | 0.02 | - | - | 0 | 0.00 | - | - | 0 |
| Spain | 0.15 | 0.14 | 0.00 | 2 | 0.04 | - | - | 0 |
| Turkey | 0.90 | 0.00 | - | 1 | 0.74 | 0.00 | - | 1 |
| UK | 0.73 | 0.06 | 0.00 | 2 | 0.95 | 0.05 | - | 1 |
| USA | 0.94 | 0.03 | - | 1 | 0.95 | 0.01 | - | 1 |

The next step is to reveal the directions of bootstrap panel Granger causalities for the pairs of *economic growth-energy consumption, economic growth-international tourist arrivals, economic growth-international tourism receipts, international tourist arrivals-energy consumption and international tourism receipts-energy consumption.* The empirical results of the bootstrap panel Granger causality test for each pair of variables and the panel are reported in detail in Tables A1–A5 in the Appendix A.

In conjunction with the information upon the number of integration for the analyzed variables, this study further looks at the direction of Granger causality between economic growth and energy consumption, between economic growth and tourist arrivals, between economic growth and tourism receipts, between tourist arrivals and energy consumption, and between tourism receipts and energy consumption. Results from the bootstrap Granger causality test due to Emirmahmutoglu and Kose (2011) for each pair of variables are reported in detail in the Appendix A. Nevertheless, Table 6 shows the summary of the outcomes obtained from the bootstrap causality method. In regards to the causal relationship, we found a one-way causal relationship running from energy consumption to economic growth (energy-led growth hypothesis) in Spain, and running from economic growth to energy consumption (growth-led energy hypothesis) in China, Turkey and Germany. We found a two-way causal relationship between energy consumption and economic growth in Italy, the USA and the panel of the top 10 most-visited countries, and no causal relationship between economic growth and energy consumption in France, Mexico, Russia and the UK. Furthermore, we found unidirectional causality from tourist arrivals to economic growth (tourism-led growth hypothesis) in China, Turkey and the panel, and from economic growth to tourist arrivals (growth-led tourism hypothesis) in Russia and Spain, bidirectional causality between economic growth and tourist arrivals in Germany, and no causality between them in France, Italy, Mexico, the UK and the USA. The evidence of one-way causality running from tourist arrivals to energy consumption (tourism-led energy hypothesis) is detected for Italy, Spain, Turkey, the UK and the USA while energy-led tourism hypothesis is held for China and Mexico. Moreover, two-way Granger causality is found for the panel of the analyzed economies, and no causality is valid for France, Germany and Russia. In addition, we found the presence of unidirectional causality from tourism receipts to economic growth in China, Germany, Turkey and the USA, and the presence of growth-led tourism hypothesis for Spain and the UK, the presence of both hypothesis for the panel, and the presence of no Granger causality in France, Italy, Mexico and Russia. Lastly, there is evidence for one-way Granger causality running from tourism receipts to energy consumption (tourism-led energy hypothesis) in Turkey and the USA, and running

from energy to receipts (energy-led tourism hypothesis) in China, Mexico, Spain and the UK, evidence of two-way causality between energy and receipts for the panel of the top ten countries, and no causal relationship between them in France, Germany, Italy and Russia.

**Table 6.** Results from Emirmahmutoglu–Kose test.

| Granger Causality between Energy Consumption and Economic Growth | |
|---|---|
| Energy-led growth | Spain |
| Growth-led energy | China, Turkey and Germany |
| Bidirectional causality | Italy, the USA and the panel of the analyzed countries |
| No causality | France, Mexico, Russia and the UK |
| **Granger Causality between Tourist Arrivals and Economic Growth** | |
| Tourism-led growth, | China, Turkey and the panel of the analyzed countries |
| Growth-led tourism | Russia and Spain |
| Bidirectional causality | Germany |
| No causality | France, Italy, Mexico, the UK and the USA. |
| **Granger Causality between Energy Consumption and Tourist Arrivals** | |
| Tourism-led energy | Italy, Spain, Turkey, the UK and the USA |
| Energy-led tourism | China, Mexico |
| Bidirectional causality | The panel of the analyzed countries |
| No causality | France, Germany and Russia |
| **Granger Causality between Tourism Receipts and Economic Growth** | |
| Tourism-led growth | China, Germany, Turkey and the USA |
| Growth-led tourism hypothesis | Spain and the UK |
| Bidirectional causality | - |
| No causality | France, Italy, Mexico, Russia and the panel of the analyzed countries |
| **Granger Causality between Tourism Receipts and Energy Consumption** | |
| Tourism-led energy | Turkey and the USA, |
| Energy-led tourism | China, Mexico, Spain and the UK |
| Bidirectional causality | The panel of the analyzed countries |
| No causality | France, Germany, Italy and Russia |

Please see the Appendix A for detailed test statistics from Emirmahmutoglu–Kose Granger causality approach.

## 5. Conclusions and Policy Recommendation

Energy and tourism have become among the most important sectors of the economy in the last several decades. As economic growth is the main indicator of the economy, it is of interest for researchers to focus on the relationship between the two most important sectors and economic growth. Thus, this empirical study aims to find the directions of Granger causality among tourism receipts and tourist arrivals, energy consumption and economic growth for the top 10 most-visited countries in the world. The top countries are responsible for about half of the world is tourism receipts, world tourist arrivals, world energy consumption and world income in the last years.

By using the Emirmahmutoglu–Kose bootstrap non-causality test, we found that an energy-led growth hypothesis is present in Spain, growth-led energy is present in China, Turkey and Germany, two-way causality is supported in Italy and the USA and no causal relationship is found between growth and energy in France, Mexico, Russia and the UK. By using the data for tourist arrivals a tourism-led growth hypothesis is present in China and Turkey a growth-led tourism hypothesis is found in Russia and Spain, bidirectional causality exists between growth and tourism in Germany, no causality occurs between the variables in France, Italy, Mexico, the UK and the USA. Tourism-led energy hypothesis is detected in Italy, Spain, Turkey, the UK and the USA an energy-led tourism hypothesis is found in China and Mexico and no causality is supported in France, Germany and Russia. By using the data for tourism receipts a tourism-led growth hypothesis exists in China, Germany, Turkey and the USA a growth-led tourism hypothesis occurs in Spain and the UK, and no Granger causality is detected in France, Italy, Mexico and Russia. Tourism-led energy hypothesis is present in Turkey and the USA an energy-led tourism hypothesis is supported in China, Mexico, Spain and the UK and no causal relationship is found in France, Germany, Italy and Russia.

Our general policy implications are guidance for researchers and governments in order to build better tourism and energy strategies. From this perspective, empirical analysis like this study plays an important role for strengthening the energy-tourism-growth literature. In other words, energy consumption, tourism development and economic growth are strongly interrelated and cause one another. Therefore, policy makers of these countries should take sustainable energy and tourism policies into account for a sustainable economic growth. Similarly, the policy makers should also take a sustainable economic growth take into account for sustainable energy and tourism policies for their countries. This study reveals the need of further empirical studies using different methods on the energy consumption-tourism development-economic growth literature. These forthcoming studies will enable the policy makers and researchers to better understand the causal relationships between these variables.

**Author Contributions:** Cem Işık and Eyüp Doğan presents the basic ideas, introduction and obtains the main results in conducts, the illustration section also the whole paper; Serdar Ongan adds his contributions to the Sections 1 and 5.

**Conflicts of Interest:** The authors declare no conflicts of interest.

## Appendix

**Table A1.** Bootstrap Granger causality between energy consumption and economic growth.

|  | $k_i$ | Energy-Led Growth Hypothesis | | Growth-Led Energy Hypothesis | |
|---|---|---|---|---|---|
|  |  | Wald Test | *p*-Value | Wald Test | *p*-Value |
| China | 2 | 0.74 | 0.69 | 4.69 * | 0.10 |
| France | 2 | 1.34 | 0.51 | 0.48 | 0.79 |
| Germany | 1 | 0.05 | 0.83 | 2.92 * | 0.09 |
| Italy | 1 | 6.86 *** | 0.01 | 7.40 *** | 0.01 |
| Mexico | 1 | 1.39 | 0.24 | 0.60 | 0.44 |
| Russia | 3 | 2.48 | 0.48 | 4.58 | 0.21 |
| Spain | 1 | 6.23 *** | 0.01 | 1.83 | 0.18 |
| Turkey | 1 | 0.92 | 0.34 | 5.35 ** | 0.02 |
| UK | 3 | 2.74 | 0.43 | 4.37 | 0.22 |
| USA | 3 | 23.97 *** | 0.00 | 25.28 *** | 0.00 |
| **Panel** | **Fisher-stat** | **50.03 ***** | **0.00** | **61.56 ***** | **0.00** |

**Note**: ***, ** and * denote the statistical significance at 1%, 5% and 10% level of significance, respectively. $k_i$ is the maximum lag length.

**Table A2.** Bootstrap Granger causality between tourist arrivals and economic growth.

|  | $k_i$ | Tourism-Led Growth Hypothesis | | Growth-Led Tourism Hypothesis | |
|---|---|---|---|---|---|
|  |  | Wald Test | *p*-Value | Wald Test | *p*-Value |
| China | 1 | 5.08 ** | 0.02 | 0.02 | 0.89 |
| France | 1 | 0.18 | 0.67 | 0.20 | 0.65 |
| Germany | 1 | 7.69 *** | 0.01 | 2.83 * | 0.09 |
| Italy | 1 | 2.40 | 0.12 | 0.06 | 0.80 |
| Mexico | 2 | 1.93 | 0.38 | 2.33 | 0.31 |
| Russia | 3 | 0.55 | 0.91 | 6.34 * | 0.10 |
| Spain | 3 | 0.79 | 0.85 | 7.58 * | 0.06 |
| Turkey | 2 | 5.94 ** | 0.05 | 0.84 | 0.66 |
| UK | 3 | 4.12 | 0.25 | 2.01 | 0.57 |
| USA | 2 | 1.18 | 0.55 | 2.26 | 0.32 |
| **Panel** | **Fisher-stat** | **35.19 ***** | **0.02** | **23.31** | **0.27** |

**Note:** ***, ** and * denote the statistical significance at 1%, 5% and 10% level of significance, respectively.

**Table A3.** Bootstrap Granger causality between energy consumption and tourist arrivals.

| | $k_i$ | Tourism-Led Energy Hypothesis | | Energy-Led Tourism Hypothesis | |
|---|---|---|---|---|---|
| | | Wald Test | *p*-Value | Wald Test | *p*-Value |
| China | 2 | 2.88 | 0.24 | 6.28 ** | 0.04 |
| France | 1 | 0.15 | 0.70 | 0.75 | 0.39 |
| Germany | 1 | 0.04 | 0.84 | 0.89 | 0.35 |
| Italy | 1 | 4.25 ** | 0.04 | 0.64 | 0.42 |
| Mexico | 2 | 0.87 | 0.65 | 10.89 *** | 0.00 |
| Russia | 2 | 0.39 | 0.82 | 0.80 | 0.67 |
| Spain | 2 | 7.96 ** | 0.02 | 1.63 | 0.44 |
| Turkey | 2 | 16.62 *** | 0.00 | 0.25 | 0.88 |
| UK | 3 | 28.05 *** | 0.00 | 1.12 | 0.77 |
| USA | 1 | 4.18 ** | 0.04 | 0.98 | 0.32 |
| **Panel** | **Fisher-stat** | **67.76 ***** | **0.00** | **28.36 *** | **0.10** |

**Note**: ***, ** and * denote the statistical significance at 1%, 5% and 10% level of significance, respectively.

**Table A4.** Bootstrap Granger causality between tourism receipts and economic growth.

| | $k_i$ | Tourism-Led Growth Hypothesis | | Growth-Led Tourism Hypothesis | |
|---|---|---|---|---|---|
| | | Wald Test | *p*-Value | Wald Test | *p*-Value |
| China | 2 | 5.87 ** | 0.05 | 0.54 | 0.76 |
| France | 3 | 0.20 | 0.98 | 5.05 | 0.17 |
| Germany | 1 | 4.09 ** | 0.04 | 0.06 | 0.80 |
| Italy | 3 | 5.51 | 0.14 | 1.42 | 0.70 |
| Mexico | 1 | 0.14 | 0.70 | 2.02 | 0.16 |
| Russia | 3 | 1.62 | 0.65 | 5.05 | 0.17 |
| Spain | 3 | 5.43 | 0.14 | 36.57 *** | 0.00 |
| Turkey | 3 | 32.04 *** | 0.00 | 3.90 | 0.27 |
| UK | 1 | 0.38 | 0.54 | 2.92 * | 0.09 |
| USA | 2 | 13.10 *** | 0.00 | 4.10 | 0.13 |
| **Panel** | **Fisher-stat** | **64.89 ***** | **0.00** | **57.48 ***** | **0.00** |

**Note**: ***, ** and * denote the statistical significance at 1%, 5% and 10% level of significance, respectively.

**Table A5.** Bootstrap Granger causality between tourism receipts and energy consumption.

| | $k_i$ | Tourism-Led Energy Hypothesis | | Energy-Led Tourism Hypothesis | |
|---|---|---|---|---|---|
| | | Wald Test | *p*-Value | Wald Test | *p*-Value |
| China | 2 | 0.42 | 0.81 | 5.34 * | 0.07 |
| France | 3 | 4.13 | 0.25 | 1.41 | 0.70 |
| Germany | 2 | 2.57 | 0.28 | 0.57 | 0.75 |
| Italy | 3 | 3.84 | 0.28 | 3.25 | 0.35 |
| Mexico | 2 | 0.75 | 0.69 | 6.55 ** | 0.04 |
| Russia | 1 | 0.08 | 0.78 | 0.02 | 0.89 |
| Spain | 3 | 4.29 | 0.23 | 7.37* | 0.06 |
| Turkey | 3 | 20.69 *** | 0.00 | 1.38 | 0.71 |
| UK | 3 | 5.33 | 0.15 | 12.13 *** | 0.01 |
| USA | 1 | 10.08 *** | 0.00 | 1.11 | 0.29 |
| **Panel** | **Fisher-stat** | **47.32 ***** | **0.00** | **34.14 *** | **0.02** |

**Note**: ***, ** and * denote the statistical significance at 1%, 5% and 10% level of significance, respectively.

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
