# Peer review of "Analyzing the Tourism–Energy–Growth Nexus for the Top 10 Most-Visited Countries"

_economies, doi:10.3390/economies5040040_

Reviewer 1 Report

The exercise is interesting from the point of view of the model used, but absolutely without any usefulness for strategies and planning. Many aspects are completely missed: first of all tourism is a demand driven sector  and cannot been explained just in relation to energy .  In terms of applied research, the issue seems, as it has been described, a non sense. 

Author Response

We thank the reviewer for his/her comments, observations and suggestions. We have now revised the manuscript closely following the reviewer’s comments and recommendations.

Reviewer 2 Report

1.     It is recommended to draw a conclusion after Table 2 in Chapter 2 (Literature Review)

2.     In Chapter 4 (Methods and Empirical Results), it is recommended to develop appropriate interpretations between Table 5 and Table 6 (a quality article includes tables or graphs with appropriate interpretations between them)

3.     In the "Conclusions" it is necessary to be presented more clearly the proposals for future research

Author Response

1.     It is recommended to draw a conclusion after Table 2 in Chapter 2 (Literature Review)

We thank the reviewer for his/her comments, observations and suggestions. We have now revised the literature part. We collect the Table 1 and Table 2 to one Table. 

2.     In Chapter 4 (Methods and Empirical Results), it is recommended to develop appropriate interpretations between Table 5 and Table 6 (a quality article includes tables or graphs with appropriate interpretations between them)

We thank the reviewer for his/her observations and providing invaluable feedback.

There are many more studies that outline the interrelationship between the energy and tourism. However, we seem to have overlooked at some of the original studies. We thank the reviewer for this remarkable observation. 

3.     In the "Conclusions" it is necessary to be presented more clearly the proposals for future research

Lines 233 and 241. The changes are also highlighted in red within the manuscript. 

Reviewer 3 Report

In the introduction, the idea of considering the 10 most visited nations appears without first having contextualized. 

In Table 1, the form of GDP measurement is not indicated.

In the introduction the origin of the study is not clear, it is necessary to mention theoretical referents. A good part of the works that it cites study a relation between the variables much more congruent. The proposed relationship is likely to have not been studied earlier, because of the lack of relevance.

In Table 2 I suggest modifying destination by space. That table 2 and the 3 I suggest organizing it chronological form. In table 2 it did not define what is RELC and GR, as well as NRELC.

You should reduce the size of Tables 2 and 3 to focus on explaining the major results and limitations of the previous work, as well as to highlight the contribution to the existing literature.

I did not find references from 2016 and 2017. I discovered that his true reference is Tang et al (2016) and another study in the part where it establishes the model, speak more of these works and indicate how you overcome them or how they helped you.

It is very important, for the purpose of replicability, to indicate the software used and to arrangement of users the database and routine or file of the base program.

For the Mexican case, where in the last 30 years the growth has been very low, its result of non-causality my seems very accurate. I ask you to quote the following work: https://www.mdpi.com/2227-7072/4/2/6

It is vital that you create a section for describing your variables of interest: energy, tourism, and GDP.

I propose that the results be limited to the relations between energy and growth and between tourism and growth, since doing so between tourism and energy seems to me it is not part of your goal.

In the results section emphasize and distinguish between the results for the first panel and the results for the country after.

In the tables in the annex, for readers, define what is meant by Ki. And as for method, explain step-by-step what you did and how you did it so that other colleagues and students can replicate it.

In the policy part, I found that it is not related to the results, it should be results and not invent without sustenance.

Check the references, I found error. I appreciate your work, it has been very useful read it.

Author Response

In the introduction, the idea of considering the 10 most visited nations appears without first having contextualized. 

We thank the reviewer for his/her comments, observations and suggestions. We have now revised it. 

In Table 1, the form of GDP measurement is not indicated.

We thank the reviewer for his/her comments, observations and suggestions. We have now revised it. (Table 1)

In the introduction the origin of the study is not clear, it is necessary to mention theoretical referents. A good part of the works that it cites study a relation between the variables much more congruent. The proposed relationship is likely to have not been studied earlier, because of the lack of relevance.

We thank the reviewer for his/her observations and providing invaluable feedback.

Lines 68 and 74. The changes are also highlighted in red within the manuscript. 

In Table 2 I suggest modifying destination by space. That table 2 and the 3 I suggest organizing it chronological form. In table 2 it did not define what is RELC and GR, as well as NRELC.

We thank the reviewer for his/her comments, observations and suggestions.

We collect Table 2 and Table 3 to one Table. We have now revised it according to suggestion. 

You should reduce the size of Tables 2 and 3 to focus on explaining the major results and limitations of the previous work, as well as to highlight the contribution to the existing literature.

We thank the reviewer for his/her comments, observations and suggestions.

We reduced them to on Table.

I did not find references from 2016 and 2017. I discovered that his true reference is Tang et al (2016) and another study in the part where it establishes the model, speak more of these works and indicate how you overcome them or how they helped you.

We thank the reviewer for his/her comments, observations and suggestions.

We added current literature to paper. Please check the Table 2. 

It is very important, for the purpose of replicability, to indicate the software used and to arrangement of users the database and routine or file of the base program.

We thank the reviewer for his/her observations and providing invaluable feedback.

Line 125. The changes are also highlighted in red within the manuscript

For the Mexican case, where in the last 30 years the growth has been very low, its result of non-causality my seems very accurate. I ask you to quote the following work: https://www.mdpi.com/2227-7072/4/2/6

We thank the reviewer for his/her observations and providing invaluable feedback.

In this study, the causal link for each country is separately examined because countries in a panel may have idiosyncratic characteristics and hence the causality may be different for each country in the panel. Although the relationship between renewable energy consumption and economic growth may not necessarily be two-sided. The theoretical framework and the existing literature showed mixed evidence supporting, one-sided, two-sided, and no relationships. Causality studies are necessary to determine the magnitude and the direction of this relationship. We conducted our analyses following the extant literature and most advanced empirical techniques to estimate the causal relationship between the study variables and provide policy recommendations.

It is vital that you create a section for describing your variables of interest: energy, tourism, and GDP.

We thank the reviewer for his/her observations and providing invaluable feedback.

Lines 21 and 25, and 11 and 120. The changes are also highlighted in red within the manuscript.

I propose that the results be limited to the relations between energy and growth and between tourism and growth, since doing so between tourism and energy seems to me it is not part of your goal.

We thank the reviewer for his/her observations and providing invaluable feedback.

Lines 21 and 25, and 11 and 120. The changes are also highlighted in red within the manuscript.

In the results section emphasize and distinguish between the results for the first panel and the results for the country after.

We thank the reviewer for his/her observations and providing invaluable feedback.

Lines 171 and 175. The changes are also highlighted in red within the manuscript

In the tables in the annex, for readers, define what is meant by Ki. And as for method, explain step-by-step what you did and how you did it so that other colleagues and students can replicate it.

We thank the reviewer for his/her observations and providing invaluable feedback.

k is the maximum lag length. We added it to end of the Table A1. 

In the policy part, I found that it is not related to the results, it should be results and not invent without sustenance.

We thank the reviewer for his/her observations and providing invaluable feedback.

Lines 233 and 241. The changes are also highlighted in red within the manuscript

Check the references, I found error. I appreciate your work, it has been very useful read it.

We thank the reviewer for his/her observations and providing invaluable feedback.

The changes are also highlighted in red within the references.
